# Effects of RAGE Deletion on the Cardiac Transcriptome during Aging

**DOI:** 10.3390/ijms231911130

**Published:** 2022-09-22

**Authors:** Francesco Scavello, Luca Piacentini, Stefania Castiglione, Filippo Zeni, Federica Macrì, Manuel Casaburo, Maria Cristina Vinci, Gualtiero I. Colombo, Angela Raucci

**Affiliations:** 1Unit of Experimental Cardio-Oncology and Cardiovascular Aging, Centro Cardiologico Monzino IRCCS, 20138 Milan, Italy; 2Bioinformatics and Artificial Intelligence Facility, Centro Cardiologico Monzino IRCCS, 20138 Milan, Italy; 3Animal Facility, Centro Cardiologico Monzino IRCCS, 20138 Milan, Italy; 4Vascular Biology and Regenerative Medicine Unit, Centro Cardiologico Monzino IRCCS, 20138 Milan, Italy; 5Unit of Immunology and Functional Genomics, Centro Cardiologico Monzino IRCCS, 20138 Milan, Italy

**Keywords:** fibrosis, adaptive immunity, cardioprotection, RAGE isoforms, transcriptome profiling

## Abstract

Cardiac aging is characterized by increased cardiomyocyte hypertrophy, myocardial stiffness, and fibrosis, which enhance cardiovascular risk. The receptor for advanced glycation end-products (RAGE) is involved in several age-related diseases. RAGE knockout (*Rage−/−*) mice show an acceleration of cardiac dimension changes and interstitial fibrosis with aging. This study identifies the age-associated cardiac gene expression signature induced by RAGE deletion. We analyzed the left ventricle transcriptome of 2.5-(Young), 12-(Middle age, MA), and 21-(Old) months-old female *Rage−/−* and C57BL/6N (WT) mice. By comparing Young, MA, and Old *Rage−/−* versus age-matched WT mice, we identified 122, 192, and 12 differently expressed genes, respectively. Functional inference analysis showed that RAGE deletion is associated with: (i) down-regulation of genes involved in antigen processing and presentation of exogenous antigen, adaptive immune response, and cellular responses to interferon beta and gamma in Young animals; (ii) up-regulation of genes related to fatty acid oxidation, cardiac structure remodeling and cellular response to hypoxia in MA mice; (iii) up-regulation of few genes belonging to complement activation and triglyceride biosynthetic process in Old animals. Our findings show that the age-dependent cardiac phenotype of *Rage−/−* mice is associated with alterations of genes related to adaptive immunity and cardiac stress pathways.

## 1. Introduction

Aging is the major risk factor for cardiovascular diseases (CVD). During aging, the heart undergoes a continuous and unavoidable tissue remodeling, characterized by an increase in cardiomyocytes hypertrophy and perivascular/interstitial fibrosis, which underlies myocardial stiffness and reduced contractile function, eventually leading to heart failure (HF) [1,2,3]. Identification of molecular mechanisms associated with age-induced cardiac changes will be useful in developing strategies aimed at extending a healthy lifespan.

The receptor for advanced glycation end-products (RAGE) is a pattern recognition receptor (PRR) expressed at a very low level in most tissues during homeostasis, except the lung [4,5,6,7]. The membrane-bound full-length RAGE (FL-RAGE) recognizes several inflammatory molecules, including advanced glycation end-products (AGEs), S100/calgranulins proteins, and high mobility group box one (HMGB1) [8,9,10,11], is rapidly up-regulated at the site of injury where its ligands accumulate and promotes the development of numerous inflammatory diseases [10,11,12,13,14]. FL-RAGE/ligands binding regulates cell migration, adhesion, and inflammation through the activation of various signaling pathways [9,15,16,17]. Soluble forms of RAGE, collectively named sRAGE, have been identified in the bloodstream and tissues of humans and rodents; they act as protective anti-inflammatory decoy receptors by preventing FL-RAGE activation [4,7,18,19,20,21]. Consistently, animals treated with sRAGE or RAGE knockout (*Rage−/−*) mice exhibit reduced tissue damage in various diseases, such as diabetes, atherosclerosis, arterial injury, and ischemic cardiac damage [7,10,11,22,23,24]. In these settings, RAGE ablation leads to reduced inflammatory cells recruitment to the site of damage and, thereby, tissue inflammation.

Conversely, recent findings highlighted that *Rage−/−* mice progress to a pathological phenotype with aging: indeed, they spontaneously develop pulmonary fibrosis and eventually cancer because of impaired DNA repair [25,26]. Accordingly, we have published that female *Rage−/−* mice exhibit an exacerbation of age-dependent left ventricle (LV) remodeling consisting of increased diastolic and systolic dimensions, enhanced interstitial fibrosis, and signs of HF onset. RAGE ablation promotes cardiac fibroblast differentiation in myofibroblasts and higher collagen fibers deposition through the activation of the profibrotic Transforming Growth Factor (TGF)-β1 pathway [7].

Nevertheless, a comprehensive molecular landscape describing the mechanism influenced by RAGE during cardiac aging is still lacking. Molecular profiling of cells and tissues represents an effective tool to identify specific signatures, which can help to understand the underlying pathological phenotype and, ultimately, develop new working hypotheses. In this study, we tested whether the absence of RAGE induces significant transcriptional changes at the cardiac level during physiological aging. To this aim, we investigated through a genome-wide approach the LV transcriptome of *Rage−/−* and control mice at different ages. Based on the results obtained, we provided inferences on the functional association between gene expression signatures differentiating *Rage−/−* from wild-type (WT) mice and the underlying age-related pathogenic mechanisms.

## 2. Results

### 2.1. Gene Expression Dataset

We analyzed the LV transcriptome of WT and *Rage−/−* female mice of three different ages: 2.5-(Young), 12-(Middle Age; MA), and 21-months-old (Old). Following the probe-filtering criteria, we identified 13277 expressed transcripts corresponding to 10174 unique genes (see annotation details in Appendix A). By using the entire adjusted probe-expression matrix, we obtained an overview of the expression data and inspected possible sub-sample grouping. As expected, the scatterplot of the two first principal components of the Principal Component Analysis (PCA), which together explain 40% of the variance, showed a clear clustering of mice according to age, i.e., the main phenotype of the dataset (Figure 1).

### 2.2. Differential Expression Analysis between Rage−/− and WT Mice

Differential expression (DE) analysis adjusted for confounders (i.e., latent variables) allowed us to unveil specific changes between *Rage−/−* vs. WT mice phenotypes at different ages. We reported the overall results and statistics for all three age comparisons in Appendix A. The main findings are summarized in Figure 2. We found 122, 192, and 12 significant DE genes (DEG) comparing *Rage−/−* vs. WT mice in Young, MA, and Old groups, respectively, suggesting that the most profound differences occur in Young and MA mice as opposed to Old mice. Changes in gene expression were technically validated by reverse-transcription quantitative PCR (RT-qPCR) on 10 selected genes that were DE in at least one of the comparisons between *Rage−/−* vs. WT in Young, MA, or Old mice. The correlation analysis, which included genes spanning from low to medium-high abundance expression levels, showed a highly significant Pearson’s coefficient (r = 0.92, *p*-Value = 8.1 × 10^−6^; see Appendix A).

### 2.3. Functional Inferences from Genome-Wide Differential Expression Analysis

To infer from the DE analysis specific biological functions characterizing *Rage−/−* and WT phenotypes over the course of mouse age, we performed a two-step procedure: (i) a hierarchical clustering to identify homogeneous groups of DEG in Young, MA, and Old mouse groups and (ii) a gene set enrichment analysis performed on the clusters of transcripts identified for each age.

*(i) Hierarchical clustering of DEG.* We performed hierarchical clustering analyses based on the expression values of DEG found comparing *Rage−/−* vs. WT mice in each age group, including the expression of those same genes in the other two age groups. We then partitioned the transcript clustering dendrograms to generate a maximum of three different transcript groups for each comparison based on the similarity of gene expression (Figure 3).

To better represent the trend over time of the gene clusters in both *Rage−/−* and WT mice age groups, we summarized each cluster’s profile by averaging the expression values of their component transcripts and used them to plot time-series graphs (Figure 4).

*(ii). Gene set enrichment analysis.* Since gene set enrichment analysis is performed at a gene level, the transcripts of each cluster were collapsed to unique gene identifiers. We tested which Gene Ontology (GO) Biological Processes (BP) and pathways were associated with each cluster of genes distinguishing the *Rage−/−* vs. WT phenotypes in Young, MA, and Old mice. Overall, we found a considerable number of significant GO-BP/pathways that were over-represented in the aforementioned gene clusters (Appendix A). We present herein the main findings for each different age group.

*(a) Young*. We observed a total of 2, 78, and 17 GO-BP/pathways associated with Cluster 1A, 2A, and 3A, respectively. The most relevant GO-BP/pathways recall mechanisms that involve antigen processing and presentation of exogenous antigen (Cluster 1A), adaptive and innate immune response functions, and cytokine, type I- and gamma-interferon mediated signaling pathways (Cluster 2A). No significant GO-BP/pathways were found in Cluster 3A (for details, see Appendix A).

*(b) MA*. Five, 31, and 43 gene sets were associated with Cluster 1B, 2B, and 3B, respectively. Relevant GO-BP/pathways included positive regulation of NIK/NF-kappaB signaling, cellular response to hypoxia, and regulation of RUNX3 for Cluster 1B; adenylate cyclase-inhibiting G protein-coupled receptor signaling pathway, oxidoreductase activity, apoptosis, and epigenetic regulation for Cluster 2B; alcohol and long-chain fatty acid metabolic process, ceramide biosynthetic process, and negative regulation of cytokine production involved in inflammatory response for Cluster 3B.

*(c) Old*. The very low number of genes constituting the three clusters did not allow to retrieve significant GO-BP/pathways, although interesting associations with single genes, including triglyceride biosynthetic process (Cluster 1C), positive regulation of phospholipid biosynthetic process (Cluster 2C), and complement activation (Cluster 3C), were observed (for details see Appendix A).

Finally, to facilitate the overall data interpretation and visualize the relationships among the most significant GO-BP/pathway gene sets, we summarized the functional enrichment analysis results into enrichment networks by high-lightening overview gene-set terms (which recapitulate closely related gene sets) of the most significant GO-BP/pathways for each cluster of genes (Figure 5 and Figure 6).

## 3. Discussion

The role of RAGE during physiological cardiac aging is still poorly understood. We have recently published that, while aging, female *Rage−/−* mice show an exacerbation of the myocardial remodeling, consisting in an increase in LV end-diastolic and -systolic dimensions and enhanced fibrosis, compared to sex- and age-matched WT animals [7]. Of note, the major differences, both with aging and between the two genotypes, occur during the Young-MA transition, while there are no major differences among the Old groups. MA *Rage−/−* mice show higher collagen fibers deposition and a larger number of activated cardiac fibroblasts (myofibroblasts) along with increased expression of profibrotic Transforming Growth Factor (TGF)-β1 pathway components compared to MA WT mice, and almost similar to the Old groups [7]. In this study, we attempted to elucidate the underlying mechanisms of accelerated age-dependent cardiac remodeling in *Rage−/−* mice through a genome-wide approach by analyzing the LV transcriptome of Young, MA, and Old *Rage−/−* and WT mice. We found that specific gene expression patterns differentiate *Rage−/−* from WT mice and, consistent with their cardiac phenotype, the most prominent gene number variations are associated with Young and MA groups (Figure 2).

In Clusters 1A and 2A, 106 out of 122 modulated transcripts were significantly under-expressed in Young *Rage−/−* animals in comparison with the WT counterpart, which did not differ among MA and Old groups (Figure 3 and Figure 4). Cluster 1A contains genes, such as intercellular adhesion molecule (*Icam1*), involved in interactions between a Lymphoid and a non-Lymphoid cell (Figure 5, Appendix A). ICAM1 is expressed on the endothelium and immune cells and promotes leukocyte recruitment during inflammation [27]. Cluster 2A contains the highest number of DEG among the two genotypes and is enriched in functional pathways such as defense response to other organisms, response to interferon-beta, cytokine-mediated signaling pathway, interferon-gamma-mediated signaling pathway, and regulation of adaptive immune response (CD4(+) and CD8(+) alpha-beta T cells activation) (Figure 5, Appendix A), suggesting that the adaptive immunity is altered by RAGE deletion in Young animals. Recent evidence shows that non-infectious myocardial injuries mobilize adaptive immunity against cardiac auto-antigens to modulate inflammation and fibrosis; furthermore, the healthy young murine myocardium harbors tissue-resident immune cells, including T lymphocytes [28,29,30]. T cell phenotypes can influence tissue remodeling in different ways depending on the specific context. CD4(+) T cells have been reported to exert an anti-fibrotic effect throughout IFN-γ secretion in experimental models of kidney and pulmonary fibrosis [31]. CD4(+) T cells participate in cardiac healing after myocardial infarction but can also aggravate heart failure induced by pressure overload [28,32,33]. Old CD4(+) T-cell-deficient mice exhibit attenuated cardiac inflammation and dysfunction but similar levels of fibrosis compared to WT mice of the same age [29,34]. Senescent T cells are more pro-inflammatory and, when injected into young lymphocyte-deficient mice, show higher cardiac tropism than young T cells; however, they are not sufficient to cause major functional and structural cardiac alterations [29,34]. Immune cells are the major source of both IFN-γ and IFN-β; however, the latter can also be produced by non-immune cells, including cardiac cells [35,36]. IFN-β has a well-established protective role in the heart; indeed, myocardial lesions induced by different types of viruses can be reduced by IFN-β treatment [37,38]. In sterile conditions, IFN-γ has been shown to exert both adverse or protective effects on the heart [36]. Of note, IFN-γ-deficient mice subject to transverse aortic constriction developed more severe cardiac hypertrophy, fibrosis, and dysfunction [39]. In vitro, IFN-γ can inhibit proliferation and the expression of profibrotic genes in human cardiac myofibroblasts [40]. Similarly, IFN-β can counteract the profibrotic activity of TGF-β1 through STAT1 and STAT2 activation in cardiac fibroblasts [35]. Herein, we show that physiological cardiac aging in WT animals is associated with the down-regulation of T-cell activation and IFN signaling pathways, and their earlier impairment, observed in Young *Rage−/−* animals, causes an acceleration of age-dependent cardiac remodeling, in particular fibrosis, suggesting a protective function of T cell-mediated type I response during heart aging.

*Rage−/−* mice lack both RAGE isoforms, i.e., FL-RAGE and sRAGE, and cardiac transcriptional profile, and thereby the phenotype of these animals likely relies on the deficiency of both proteins. It is well known that FL-RAGE plays a primary role in regulating the inflammatory response associated with a wide range of diseases. Specifically, the adhesive and pro-migratory functions of FL-RAGE are crucial for leukocyte recruitment to injured tissues even in cooperation with ICAM1 [9,14,41,42]. Furthermore, FL-RAGE activation is specifically required for T cell expansion and Th1 CD4(+) differentiation, and *Rage*-deficient T cells display reduced IFN-γ production [43,44]. As far as it concerns the soluble isoforms, we published that circulating sRAGE acts as a cardioprotective molecule with anti-fibrotic activity; its levels decline with aging in healthy humans and mice, and administration of the recombinant protein in MA animals reduces age-dependent cardiac fibrosis [7,20,21]. Interestingly, the latest studies demonstrate that sRAGE can induce IFN-γ to protect the heart from injuries [45,46]. Hence, the absence of both FL-RAGE and sRAGE may be responsible for the impaired T cell-mediated type I response of Young *Rage−/−*.

In line with Young groups data, in the cluster 1B of MA animals, which includes transcripts under-expressed in MA *Rage−/−* compared to the WT group, we found genes related to the regulation of expression and activity of Runt-Related Transcription Factor 3 (RUNX3; Figure 3, Figure 4 and Figure 6), that is a crucial factor for T cells differentiation and IFN-γ production [47,48]. These data further corroborate that the down-regulation of the IFN pathway may be one of the principal causes of cardiac vulnerability and adverse age-related cardiac remodeling of *Rage−/−* animals. Moreover, the RUNX3 pathway includes genes, such as *Psmb7* and *Ep300*, that act as protective factors in response to cardiac oxidative stress (Appendix A; [49,50,51]), suggesting an impaired cardioprotective response in MA *Rage−/−* mice compared to WT counterpart.

Signs of cardiac stress in MA *Rage−/−* mice are also evidenced by the increased expression of genes related to energy metabolism. Indeed, cluster 3B includes metabolic processes such as alcohol metabolic and biosynthetic processes, long-chain fatty acid metabolic process, and ceramide biosynthetic process (Figure 6). The aged heart accumulates long-chain fatty acids incorporated in triglycerides, phospholipids, and other lipid subspecies, including ceramides [52]. Lipid peroxidation increases in the aged heart in response to oxidative stress as a mark of adverse cardiac remodeling [52]. Ceramides, molecules constituted by sphingosine bound to fatty acyl chains of different lengths, are able to activate several signal transduction pathways, and their accumulation is strongly associated with the pathogenesis of many CVD [53]. *Sphk1* and *Ephx2* genes are central to many of the biological networks of cluster 3B and are overexpressed in MA *Rage−/−* mice (Figure 6, Appendix A). The sphingosine kinase type 1 (SPHK1) phosphorylates sphingosine leading to the formation of sphingosine-1-phosphate (S1P), a potent signaling lipid with profibrotic actions [54]. It has been demonstrated that TGF-β1 induces the expression of SPHK1 to stimulate collagen production in both cardiac fibroblasts and myofibroblasts through S1P signaling [54,55]. The *Ephx2* gene encodes for soluble epoxide hydrolase (sEH) involved in the metabolism of arachidonic acid, a polyunsaturated fatty acid released into the cytosol in response to cardiac stressors [56]. sEH genetic ablation and pharmacological inhibition have been shown to exert cardioprotective actions in myocardial infarction and HF, improving mitochondrial function, reducing oxidative stress and inflammation, and opposing apoptosis [56]. Although RAGE activation has been associated with lipid accumulation and peroxidation in several organs [57,58,59], its specific role in intramyocardial lipid accumulation remains unknown. Lipid peroxidation is an alternative route for Nε-(carboxymethyl)lysine (CML) formation, which is a major AGE also involved in cardiac fibrosis [60,61]. Our data suggest that RAGE deletion affects cardiac fatty acids accumulation/oxidation in MA mice, likely, because of sRAGE absence, which may protect against CML formation and activity.

A corroboration that *Rage−/−* mice present a stressed cardiac phenotype is also given by the presence of GO categories related to the regulation of DNA damage response and signal transduction by p53 class mediators (Figure 6). Finally, according to the fibrotic phenotype of MA *Rage−/−* mice, the enrichment analysis of cluster 3B also evidenced functional pathways such as positive regulation of collagen biosynthetic process and R-SMAD binding (Figure 6; [62,63]). Of note, MA *Rage−/−* mice evidence a transcriptional drift of genes related to the regulation of cardiac cell growth and hypertrophy (Cluster 2B, Figure 6): some of them are positive regulators of cardiac hypertrophy, such as *Edn1*, while others, such as *Ifngr2-Naftcs*, mediate antihypertrophic effects (Appendix A; [64,65]). We published that *Rage−/−* mice do not exhibit altered cardiomyocyte hypertrophy when compared to control mice at any age [7], suggesting that these animals have developed compensatory mechanisms likely due to the differential modulation of genes regulating cardiac muscle hypertrophy.

Hence, we found that RAGE deletion in mice induces alterations in LV transcriptome that affect cardiac remodeling (i.e., fibrosis) during aging. LV of Young *Rage−/−* mice exhibits the down-regulation of genes related to adaptive immunity, T-cell differentiation and activation, and IFN signaling pathways that, eventually, accelerate the development of an age-associated profibrotic phenotype. MA *Rage−/−* mice are characterized by an increase in cardiac stress marker genes associated with fatty acid oxidation, cardiac structure remodeling, impaired response to hypoxia, and programmed cell death pathways. 

This study has some limitations that require further investigation. First, a more in-depth exploration of the identified DEG and pathways is needed to confirm their involvement in the age-associated cardiac phenotype of *Rage−/−* mice. Second, it is known that gender differences are widespread in the onset and development of CVD, so their impact should be considered for CVD prevention, diagnosis, and treatment. In the present study, we used only female mice, but we are conducting similar experiments in male animals to highlight whether RAGE could influence gender differences during cardiac aging. Third, it would be helpful to discriminate the specific contribution of each RAGE isoform (FL-RAGE and sRAGE) to the aforementioned pathways during cardiac aging to develop possible new intervention strategies aimed at prolonging a healthy lifespan.

## 4. Material and Methods

### 4.1. Animals

Female C57BL/6N (WT; Charles River Laboratories, Calco, Italy) or *Rage−/−* mice were housed in standard cages and fed a normal chow diet. Ten-week-, twelve-, or twenty-one-month-old (Young, MA, and Old, respectively) animals were generated [7], weighted, and anesthetized with an intraperitoneal (i.p.) injection of ketamine-medetomidine cocktail (100 mg/Kg–10 mg/Kg). Then, mice were perfused with phosphate-buffered saline, and hearts were dissected and immediately frozen. All procedures involving animals were performed following our Institutional Guidelines, which comply with national (D.L. n.116, G.U. suppl. 40, 18 February 1992) and international laws (EU Directive 2010/63/EU). The study was authorized by the National Ministry of Health-University of Milan Committee (Approval number 12/12-30012012).

### 4.2. Microarray Gene Expression Analysis

Total RNA was extracted from the LV of mice of different age groups using miRNeasy kit (Qiagen, Hilden, Germany) and treated with DNase on a column to eliminate genomic contamination, following the manufacturer’s instructions. RNA yield/purity and RNA integrity were assessed by the Infinite M200 PRO multimode microplate reader (Tecan, Männedorf, Switzerland) and the 2100 Bioanalyzer (Agilent Technologies, Santa Clara, CA, USA), respectively.

The Illumina Total Prep RNA Amplification Kit (Life Technologies, Carlsbad, CA, USA) was used to generate labeled, linearly amplified complementary RNA (cRNA), according to the manufacturer’s manual. Briefly, 300 ng of total RNA was reverse-transcribed to cDNA using an oligo(dT) primer containing a T7 promoter sequence. Second-strand cDNA was subsequently synthesized and then in vitro transcribed, adding biotin-dNTPs. After column-based purification and ammonium acetate/ethanol precipitation, cRNA was quantified by the Infinite M200 PRO microplate reader, and the cRNA profile of all samples was checked by the RNA 6000 Nano Assay kit in an Agilent 2100 Bioanalyzer. Two μg of cRNA per sample were then hybridized at 58 °C for 18 h on MouseWG-6 v2.0 Gene Expression BeadChip (Illumina, San Diego, CA, USA), followed by detection signal reaction with the fluorolink streptavidin-Cy3 (GE Healthcare Life Sciences, Chicago, IL, USA), as recommended by manufacturer’s instructions. The iSCAN System (Illumina) was used to scan each array on the BeadChips.

Data and materials used in the current study are available in the NCBI GEO database (accession number GSE210192).

### 4.3. Reverse-Transcription Quantitative PCR (RT-qPCR)

RNA was retrotranscribed with iScript Reverse Transcription Supermix kit (#1708840, Bio-Rad, Hercules, CA, USA) following the manufacturer’s instructions. RT-qPCR was performed on a Bio-Rad iCycler Thermal Cycler (Bio-Rad, Hercules, CA, USA) with the iQ5 Multicolor Real-Time PCR Detection System, using the iQ SYBR Green Supermix (Bio-Rad, Hercules, CA, USA), specific oligos (Appendix A), and 10 ng of cDNA. Relative gene expression was determined using the 2-ΔΔCt method, normalizing to the average of 2 reference genes (*Hprt* and *Ldha*).

### 4.4. Data Processing

Array data export and quality control were performed with the Genome Studio Software v2011.1 (Illumina). Raw data were imported into the R software v4.0.2 and normalized with the Lumi R/Bioconductor package [66].

Data variance stabilization was performed by variance stabilizing transformation (VST), and transformed data were normalized by the robust spline normalization (RSN) algorithm, which combines the features of quantile and loess normalization. Microarray probes were annotated through the lumiMouseIDMapping R/Bioconductor package [67]. For subsequent analysis, we retained probes with a detection *p* value < 0.01 in at least 25% of samples. To control systematic heterogeneity in high-throughput data, a data adjustment step was performed through the DaMiRseq R/Bioconductor package [68]. Setting the DaMiR.SV() function with default parameters and method = ”fve”, 7 putative surrogate (also known as latent) variables (sv) were identified. The first 5 surrogate variables, which were associated with technical variability, were subsequently included in the statistical model design to adjust the differential expression analysis.

### 4.5. Statistical Analysis

All the statistical analyses were performed in the R environment v4.0.2. The *plotPCA()* function of the DESeq2 R/Bioconductor package was used to run Principal Component Analysis (PCA) [69]. Differential expression analysis was performed through the limma R/Bioconductor package [70]. We deemed genes as significantly different at a false discovery rate adjusted *p*-value < 0.05. An additive linear model for a multilevel experiment, which includes mouse age (i.e., Young, Middle-Age, and Old) and genotype (i.e., *Rage−/−* and Wild Type) factors, was designed. The implementation of this statistical model, adjusted for the 5 surrogate variables, allowed computing the comparisons between *Rage−/−* vs. Wild Type in (1) Young, (2) Middle-Age, and (3) Old mice. The robustness of the differential expression analysis results was assessed by exploring the histograms of the *p*-value distribution, which showed a uniformly flat distribution across the unit interval (null P values) with a peak near zero (*p* values for alternative hypotheses) [71].

### 4.6. Hierarchical Clustering of Gene Expression Data

Supervised hierarchical clustering analysis was performed based on the Euclidean and Pearson’s correlation metric for samples and genes, respectively, and the average linkage method, as implemented in the GENE-E software v3.0.215 (http://www.broadinstitute.org/cancer/software/GENE-E/index.html, accessed on 22 September 2016).

### 4.7. Functional Inferences on Gene Expression Profiles

The biological functions associated with the differentially expressed genes by comparing *Rage−/−* vs. Wild Type in the three mouse ages were inferred by taking advantage of prior biological knowledge on genes grouped by gene ontology (GO) biological processes (BP) and by Reactome pathway database (http://www.reactome.org/, accessed on 8 May 2020) using the ClueGO software v2.5.7 [72] implemented as an app in the Cytoscape v.8.0.1 platform [73]. Input gene lists were loaded into the ClueGO as Entrez ID. Parameters used to run the analysis included: Enrichment = two-sided hypergeometric test; minimal number of genes = 2; *p*-Value correction method = Benjamini–Hockberg; Custom Reference Set = the genes deemed as expressed by the above-reported filtering criteria. Specific GO-BP and pathways were deemed as enriched for an adjusted-*p* Value <0.05.

## Figures and Tables

**Figure 1 ijms-23-11130-f001:**
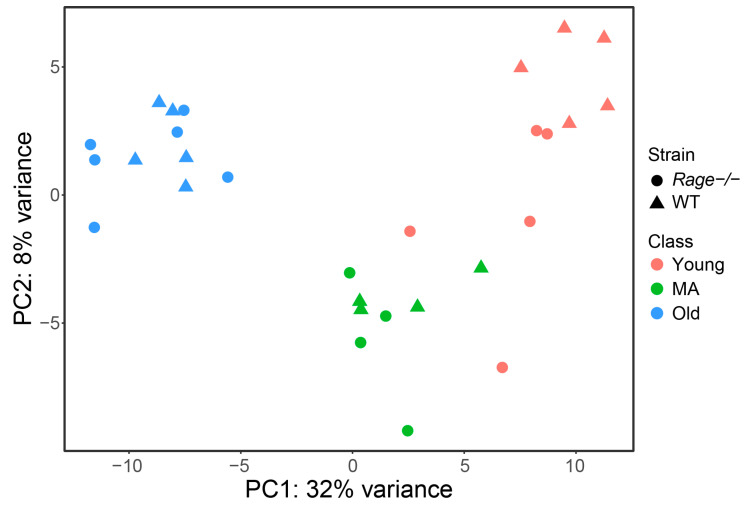
Unsupervised clustering of mice samples by Principal Component Analysis (PCA). A scatterplot of the first 2 principal components (PC1 and PC2) obtained from the PCA was performed using the whole gene-expression dataset. Colors refer to mouse age (red = Young, green = MA, blue = Old); shape represents genotypes (triangles = WT, circles = *Rage−/−*). WT = Wild Type; MA = Middle Age. Young WT n = 5, Young *Rage−/−* n = 5, MA WT n = 4, MA *Rage−/−* n = 4, Old WT n = 5, Old *Rage−/−* n = 6.

**Figure 2 ijms-23-11130-f002:**
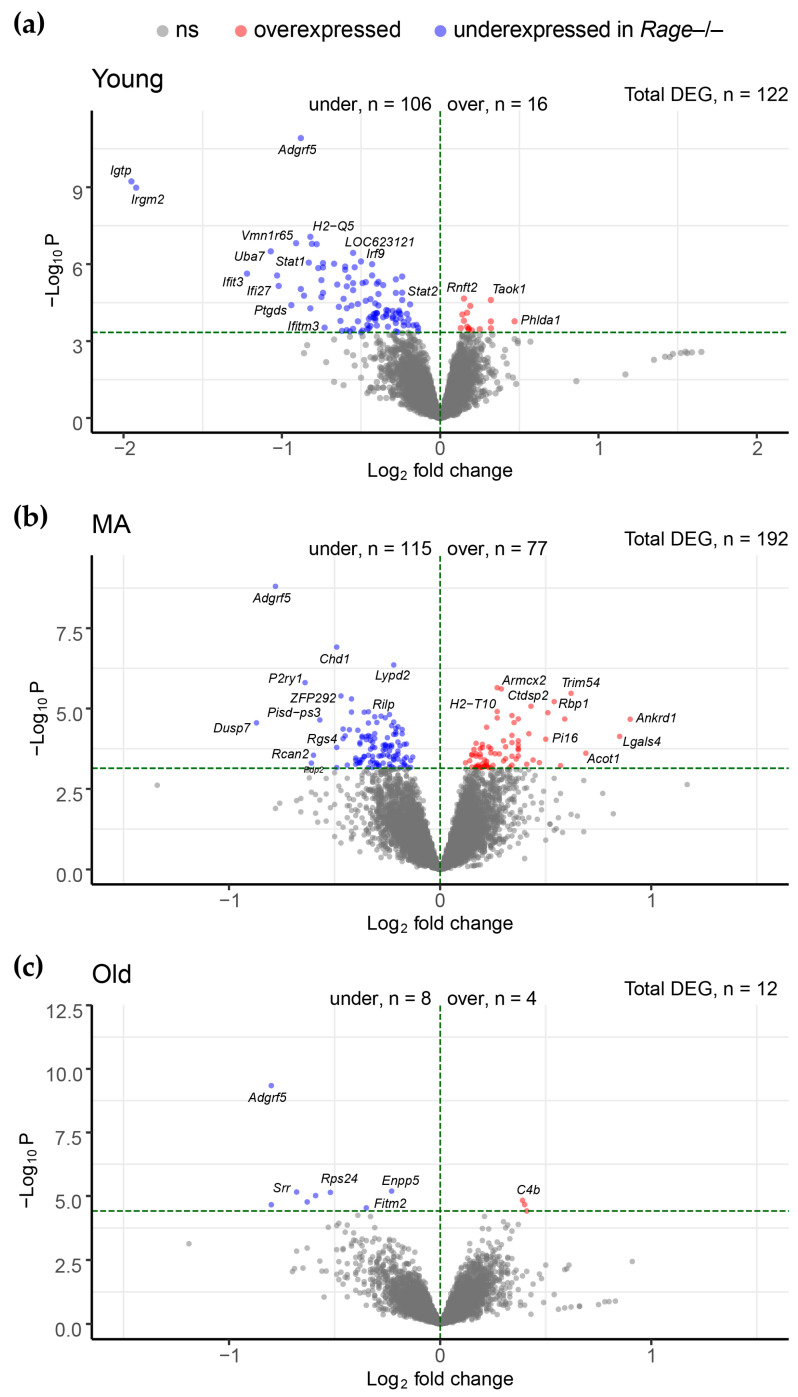
Volcano plots. Scatterplots of mean fold changes vs. significance levels (*x*- and *y*-axis, respectively) for comparisons of *Rage−/−* vs. WT in (**a**) Young, (**b**) MA and (**c**) Old mice. Red and blue dots represent significant differentially expressed genes (DEG) that stood adjustment for multiple testing (adjusted *p*-value < 0.05) and were up- and down-regulated in *Rage−/−*, respectively. Gray dots refer to non-significant DEG. A horizontal dashed line marks the threshold for -Log10 P corresponding to an adjusted *p*-Value < 0.05. −Log10 P = −Log10 nominal P-Value; Log2FC = log2 fold changes; WT = Wild Type; MA = Middle Age. Young WT n = 5, Young *Rage−/−* n = 5, MA WT n = 4, MA *Rage−/−* n = 4, Old WT n = 5, Old *Rage−/−* n = 6.

**Figure 3 ijms-23-11130-f003:**
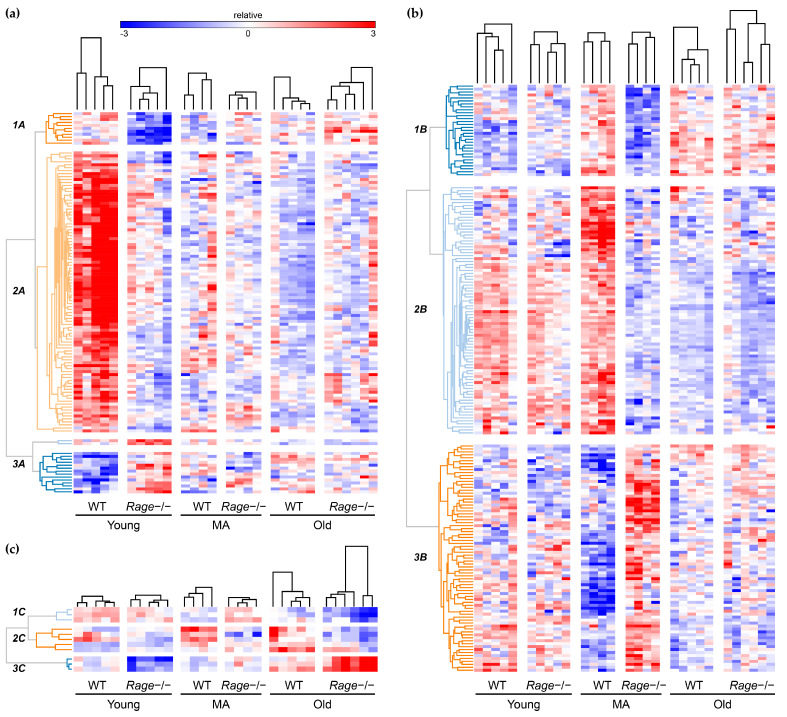
Hierarchical clustering. Hierarchical clustering of differentially expressed genes (DEG) that were identified through the comparisons between *Rage−/−* and WT in (**a**) Young, (**b**) MA, and (**c**) Old mice. Each heat map includes the expression levels of DEG in one age group and those same genes in the other two age groups. The heat maps show the level of transcripts expression from lower (dark blue) to higher (dark red). Mice (columns) were grouped based on age and genotype. Transcripts (rows) were clustered by applying an unsupervised Pearson’s correlation metric and average linkage method. WT = Wild Type; MA = Middle Age. Young WT n = 5, Young *Rage−/−* n = 5, MA WT n = 4, MA *Rage−/−* n = 4, Old WT n = 5, Old *Rage−/−* n = 6.

**Figure 4 ijms-23-11130-f004:**
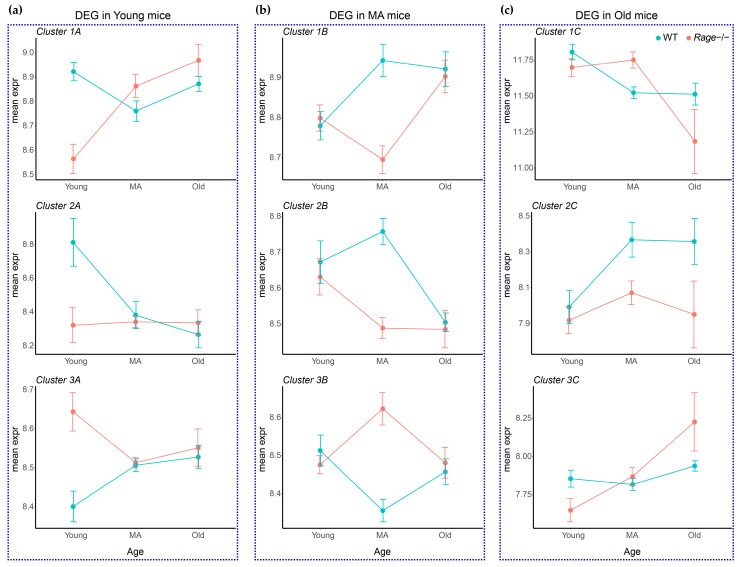
Time-series. The average expression values of the genes constituting each cluster obtained by hierarchical clustering (cf. panels (**a**) through (**c**) in Figure 3) were used to draw line plots showing the expression trend of these gene clusters. Panel (**a**) on the left represents the average expression of gene clusters 1A, 2A, and 3A in *Rage−/−* vs. WT Young mice. Panel (**b**) in the center depicts the average expression of gene clusters 1B, 2B, and 3B in *Rage−/−* vs. WT MA mice, and panel (**c**) on the right the 1C, 2C, and 3C clusters in Old mice. Pink and light blue lines and dots refer to *Rage−/−* and WT mice, respectively. Bars represent the standard deviation of the mean expression values between mice of the same age group. WT = Wild Type; MA = Middle Age. Young WT n = 5, Young *Rage−/−* n = 5, MA WT n = 4, MA *Rage−/−* n = 4, Old WT n = 5, Old *Rage−/−* n = 6.

**Figure 5 ijms-23-11130-f005:**
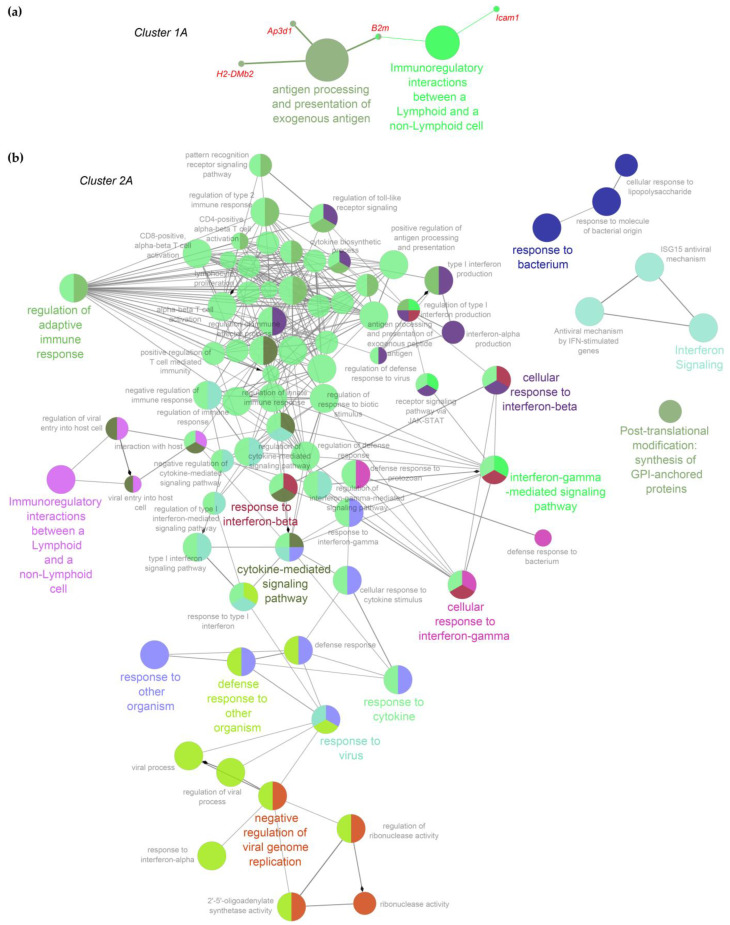
Enrichment networks in Young animals for clusters 1A (**a**) and 2A (**b**). The networks show the most significant gene ontology (GO)-biological processes (BP)/pathways (adjusted *p*-Value < 0.05) that are enriched in the transcript clusters identified by hierarchical clustering. Colors refer to groups of gene sets with related biological functions. Node size relates to gene-set significance.

**Figure 6 ijms-23-11130-f006:**
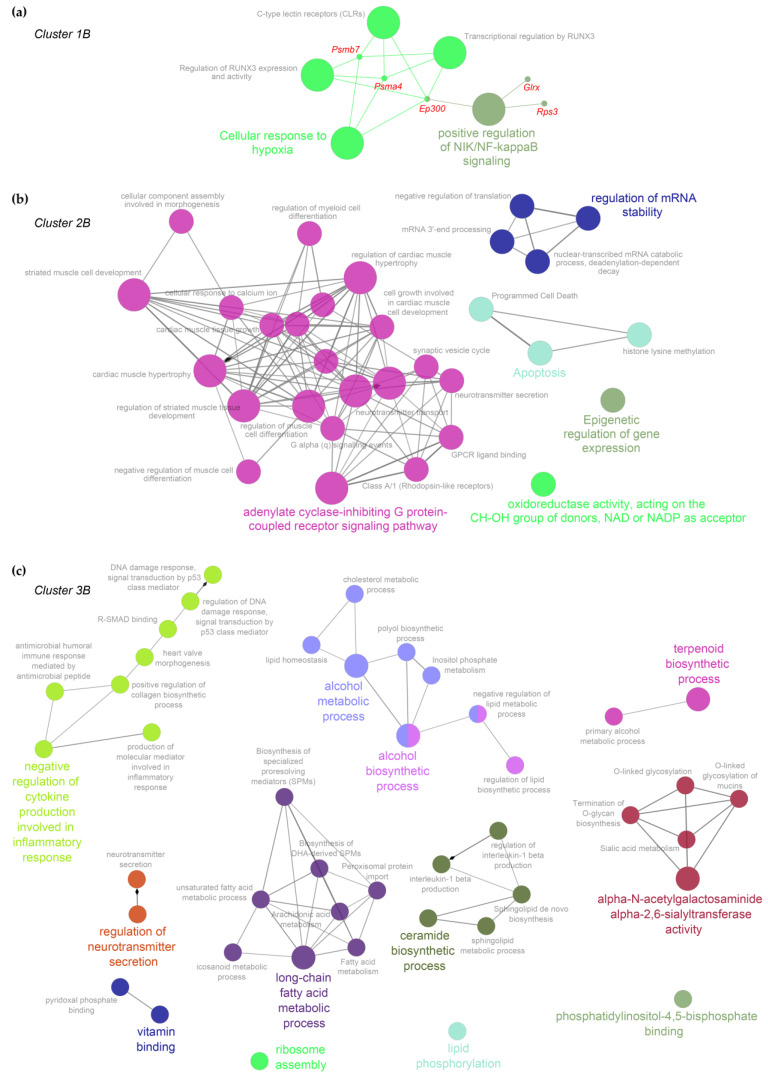
Enrichment networks in Middle Age animals for clusters 1B (**a**), 2B (**b**), and 3B (**c**). The networks show the most significant gene ontology (GO)-biological processes (BPs)/pathways (adjusted *p*-Value < 0.05) enriched in the transcript clusters identified by hierarchical clustering. Colors refer to groups of gene sets with related biological functions. Node size relates to gene-set significance.

## Data Availability

The raw and normalized microarray gene expression data used in the current study are available in the NCBI GEO database (accession number GSE210192).

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
