# Peer review of "Effects of RAGE Deletion on the Cardiac Transcriptome during Aging"

_ijms, 2022, doi:10.3390/ijms231911130_

Round 1

Reviewer 1 Report

The article, “Effects of RAGE deletion on the cardiac transcriptome during aging”, investigates Rage knockout (Rage-/-) mice for their role in cardiac associated changes related with aging. The study identifies age-associated cardiac gene expression signature in the background of RAGE deletion. It’s interesting to observe alterations of genes related to adaptive immunity and cardiac stress pathways as age-dependent cardiac phenotype of Rage-/- mice. The article is informative, and the transcriptome data is provided in nice detail. However, it would have been great to see validation of the transcriptome data. The article can be considered for publication after a revision.

A few of the comments are as follows.

1.     The information regarding specific number of mice in each group is missing. It would be better to mention number of animals in each group.

2.     The functional validation of critical genes (DEGs) from the study via RT-PCR seems missing. It would be great to add a validation of a few genes found deregulated via transcriptome analysis.

Author Response

Reviewer_1

Comments and Suggestions for Authors:

The article, “Effects of RAGE deletion on the cardiac transcriptome during aging”, investigates Rage knockout (Rage-/-) mice for their role in cardiac associated changes related with aging. The study identifies age-associated cardiac gene expression signature in the background of RAGE deletion. It’s interesting to observe alterations of genes related to adaptive immunity and cardiac stress pathways as age-dependent cardiac phenotype of Rage-/- mice. The article is informative, and the transcriptome data is provided in nice detail. However, it would have been great to see validation of the transcriptome data. The article can be considered for publication after a revision.

- We thank the Reviewer for appreciating our manuscript.

A few of the comments are as follows.

  1. The information regarding specific number of mice in each group is missing. It would be better to mention number of animals in each group.

- We thank the Reviewer for the suggestion. We have included the number of each group of mice in the figure legend (Figures 1-4) of the revised manuscript, accordingly.

  1. The functional validation of critical genes (DEGs) from the study via RT-PCR seems missing. It would be great to add a validation of a few genes found deregulated via transcriptome analysis.

- We followed the suggestion of the Reviewer and added a technical validation by RT-qPCR on 10 genes that were found differentially expressed in at least one of the comparisons between Rage-/- vs. WT in Young, MA, or Old mice as reported in the Result section (page 3, line 102-107) and the Supplementary data (Figure S1) of the revised manuscript.

Reviewer 2 Report

The authors investigated the effects of RAGE deletion on the cardiac transcriptome during aging. They showed an age-dependent cardiac phenotype resulting in downregulation and up-generation of gene involves in adaptive immunity and cardiac stress response pathways. The manuscript has a potential for high impact to the journal.

Author Response

Reviewer_2

Comments and Suggestions for Authors:

The authors investigated the effects of RAGE deletion on the cardiac transcriptome during aging. They showed an age-dependent cardiac phenotype resulting in downregulation and up-generation of gene involves in adaptive immunity and cardiac stress response pathways. The manuscript has a potential for high impact to the journal.

-We thank the Reviewer for his/her appreciation of our manuscript.

Reviewer 3 Report

I read with interest the paper entitled "Effects of RAGE deletion on the cardiac transcriptome during 2 aging" by Scavello et al. After carefull reading of the manuscript I have the following comments/suggestions/questions for the authors:

1. I miss a limitations of the study section in this paper. I would recommend to add a paragraph with potential limitations of the study.

2. It is well-known that in the clinical setting cardiovascular disease affects more males than females in general. In the current study the authors used only female mice. Do they have any evidence that the results may be different among male mice or in a mixed population of males and females? Given the fact that many differences exist in the clinical setting between males and females with cardiovascular disease which affect outcomes this issue deserves further discussion in the manuscript.

3. What are the potential clinical implications of this research? What are the future experimental and clinical steps in order to develop novel diagnostic and therapeutic interventions for the management of cardiovascular diseases?

4. The authors claim that RAGE -/- mice exhibit left ventricle remodeling through enhanced interstitial fibrosis. Did they detect any specific collagen pathway in the current study which may be involved with such a fibrosis process? 

Author Response

Reviewer_3

Comments and Suggestions for Authors:

I read with interest the paper entitled "Effects of RAGE deletion on the cardiac transcriptome during 2 aging" by Scavello et al. After carefull reading of the manuscript I have the following comments/suggestions/questions for the authors:

  1. I miss a limitations of the study section in this paper. I would recommend to add a paragraph with potential limitations of the study.

- We thank the Reviewer for the suggestion. We added a paragraph with the potential limitations in the discussion (page 12, lines 326-335 of the revised manuscript).

  1. It is well-known that in the clinical setting cardiovascular disease affects more males than females in general. In the current study the authors used only female mice. Do they have any evidence that the results may be different among male mice or in a mixed population of males and females? Given the fact that many differences exist in the clinical setting between males and females with cardiovascular disease which affect outcomes this issue deserves further discussion in the manuscript.

- The Reviewer is right. As we mentioned in the Discussion section of the revised manuscript (page 12, lines 328-332), gender differences are manifold in the development and progression of various cardiovascular diseases. Therefore, the gender impact on the prevention, diagnosis, and treatment of CVD should be considered. In order to study the contribution of RAGE to gender differences in cardiac aging, we are generating male WT and Rage-/- of different ages. Cardiac functionality and phenotype will be evaluated and compared to female counterparts. These data will be part of a new manuscript, hopefully.

  1. What are the potential clinical implications of this research? What are the future experimental and clinical steps in order to develop novel diagnostic and therapeutic interventions for the management of cardiovascular diseases?

- As we mentioned in the Discussion section, the next experimental steps will be: 1. To perform a more in-depth exploration of the identified DEG and pathways in order to confirm their involvement in the age-associated cardiac phenotype of Rage-/- mice; 2. To extend the study to male animals; 3. To discriminate the contribution of RAGE isoforms (FL-RAGE and sRAGE) to cardiac aging starting from the altered pathways identified in this study (page 12, lines 326-335 of the revised manuscript). We have already demonstrated that circulating sRAGE levels decline with aging in mice and administration of the recombinant protein in Middle age animals reduces age-dependent cardiac fibrosis, suggesting an anti-fibrotic activity of this molecule (Scavello et al 2021-2022 PMID: 34326683, 34236256, 30903794; now mentioned at pages 10-11 lines 261-263 of the revised manuscript). We also demonstrated that in healthy subjects sRAGE is a marker of aging (circulating levels of cRAGE, the most representative sRAGE isoforms in humans, decrease and negatively correlate with age, PMID: 34236256; 30903794). Clinical studies aimed at investigating the feasible value of sRAGE as a biomarker of cardiac aging and as a measure of “biological age” in humans will be very helpful. We also suggest that the preservation of appropriate levels of sRAGE in the blood may be explored as a therapeutic strategy in order to promote successful aging and increase a healthy lifespan.

  1. The authors claim that RAGE -/- mice exhibit left ventricle remodeling through enhanced interstitial fibrosis. Did they detect any specific collagen pathway in the current study which may be involved with such a fibrosis process? 

- We thank the Reviewer for this observation. Indeed, we observed a significant association with the modulation of the Gene Ontology Biological Process (GO-BP): “positive regulation of collagen biosynthetic process” and “R-SMAD binding” for the module 3B characterized by genes with higher expression levels in Rage-/- vs. WT MA mice (see Figure 6c cluster 3B and Table S2, “Middle age (MA)” sheet). This finding can support the observation of the known pro-fibrotic effect exhibited in the left ventricle by Rage-/- mice as we already discussed in the Discussion section of the manuscript (now on page 11, lines 308-311 of the revised manuscript).

Reviewer 4 Report

By this papers authors analyzed effects of RAGE deletion on mouse cardiac gene expression by means of gene expression analysis.

Overall this manuscript is interesting and data reported are clear and nicely supported by results.

Actually what is missing in my opinion is:

1. Number of young, MA and old mices should be reported since gene expression variability for each cathegory shold be taken into account. Was each cathegory of mice analyzed for gene expression variability for both RAGE depleted and WT mice? Please provide an explaination

2. Is there any sexual-dependent gene expression variability?

3. Enrichment network analysis for cluster 1C, 2C and 3C is missing. Please provide it or provide a justification in text for its absence

4. figure legend in figure 4 does not allow an immediate comprehension of the picture. Please add figure legend to the first graph of panel a, b and c

Author Response

Reviewer_4

Comments and Suggestions for Authors:

By this papers authors analyzed effects of RAGE deletion on mouse cardiac gene expression by means of gene expression analysis.

Overall this manuscript is interesting and data reported are clear and nicely supported by results.

- We thank the Reviewer for appreciating our manuscript.

Actually, what is missing in my opinion is:

  1. Number of young, MA and old mices should be reported since gene expression variability for each cathegory shold be taken into account. Was each cathegory of mice analyzed for gene expression variability for both RAGE depleted and WT mice? Please provide an explaination.

- We thank the Reviewer for this observation, which allows us to better clarify this issue. The number of mice varies from 4 to 6 per condition and we detailed the exact number of mice for each group in the Material and Methods section and in each figure legend (Figures 1-4) of the revised manuscript. Indeed, we had no reason to speculate that global gene expression variability might be specifically related to different groups of mice. Yet, our analysis pipeline included two key steps for controlling data variability: (i) the limma R/Bioconductor software package, used for differential expression analysis, implements an Empirical Bayes statistical test using moderated gene-wise variances that have been shown to be very effective in identifying actually differentially expressed genes, especially when the number of biological replicate samples is small; (ii) we estimated the presence of unwanted sources of variation by implementing a surrogate variable analysis through the DaMiRseq R/Bioconductor package and included the retrieved surrogate (aka latent) variables into the statistical model for adjusting the differential expression analysis (as described in the Material and Methods section of the manuscript). Applying these approaches, we are fairly confident that most of the effects of inter-sample variability, as well as specific (and unknown) confounding factors, have been controlled for.

For concept and method details, the Reviewer may also consult the following references:

(i)    Phipson, B, Lee, S, Majewski, IJ, Alexander, WS, and Smyth, GK (2016). Robust hyperparameter estimation protects against hypervariable genes and improves power to detect differential expression. Annals of Applied Statistics 10(2), 946–963; and

(ii)    Section 9.6.1 of the limma R/Bioconductor user guide by Gordon K. Smyth (Last revised 14 November 2021).

(iii)   Leek JT, Scharpf RB, Bravo HC, Simcha D, Langmead B, Johnson WE, Geman D, Baggerly K, Irizarry RA. Tackling the widespread and critical impact of batch effects in high-throughput data. Nat Rev Genet. 2010; 11:733–739.

(iv)   Jaffe, A. E. et al. Practical impacts of genomic data ‘cleaning’ on biological discovery using surrogate variable analysis. BMC Bioinformatics 16, 372 (2015).

  1. Is there any sexual-dependent gene expression variability?

- As we stated in the Material and Methods section, the study was conducted on female mice and, thus, it was not possible to assess any sex-dependent gene expression variability. We are aware that this is a limitation of the study and we stated this in the Discussion section of the revised manuscript (page 12, lines 326-335).

  1. Enrichment network analysis for cluster 1C, 2C and 3C is missing. Please provide it or provide a justification in text for its absence.

- The Reviewer probably missed the justification already given in the original manuscript. The reason why we did not show enrichment networks for clusters 1C, 2C, and 3C (Old mice) is reported in the Result section (now on page 7, lines 177-181 of the revised manuscript): "(c) Old. The very low number of genes constituting the three clusters did not allow to retrieve significant GO-BP/pathways although interesting associations with single genes including triglyceride biosynthetic process (Cluster 1C), positive regulation of phospholipid biosynthetic process (Cluster 2C), and complement activation (Cluster 3C) were observed (for details see Table S2)". Enrichment analysis parameters and thresholds are reported in paragraph "4.7. Functional Inferences on Gene Expression Profiles" of the Material and Methods section (now on pages 13-14, lines 411-421 of the revised manuscript).

  1. Figure legend in figure 4 does not allow an immediate comprehension of the picture. Please add figure legend to the first graph of panel a, b and c

- We have now changed the legend in Figure 4 according to the Reviewer's suggestion. We hope this caption is clear enough to help the reader understand the Figure.
